# Artificial Intelligence for Detecting and Quantifying Fatty Liver in Ultrasound Images: A Systematic Review

**DOI:** 10.3390/bioengineering9120748

**Published:** 2022-12-01

**Authors:** Fahad Muflih Alshagathrh, Mowafa Said Househ

**Affiliations:** Division of Information and Computing Technology, College of Science and Engineering, Hamad Bin Khalifa University, Qatar Foundation, Education City, Doha P.O. Box 34110, Qatar

**Keywords:** artificial intelligence, deep learning, machine learning, fatty liver, NAFLD, ultrasound

## Abstract

Background: Non-alcoholic Fatty Liver Disease (NAFLD) is growing more prevalent worldwide. Although non-invasive diagnostic approaches such as conventional ultrasonography and clinical scoring systems have been proposed as alternatives to liver biopsy, their efficacy has been called into doubt. Artificial Intelligence (AI) is now combined with traditional diagnostic processes to improve the performance of non-invasive approaches. Objective: This study explores how well various AI methods function and perform on ultrasound (US) images to diagnose and quantify non-alcoholic fatty liver disease. Methodology: A systematic review was conducted to achieve this objective. Five science bibliographic databases were searched, including PubMed, Association for Computing Machinery ACM Digital Library, Institute of Electrical and Electronics Engineers IEEE Xplore, Scopus, and Google Scholar. Only peer-reviewed English articles, conferences, theses, and book chapters were included. Data from studies were synthesized using narrative methodologies per Preferred Reporting Items for Systematic Reviews and Meta-Analyses (PRISMA) criteria. Results: Forty-nine studies were included in the systematic review. According to the qualitative analysis, AI significantly enhanced the diagnosis of NAFLD, Non-Alcoholic Steatohepatitis (NASH), and liver fibrosis. In addition, modalities, image acquisition, feature extraction and selection, data management, and classifiers were assessed and compared in terms of performance measures (i.e., accuracy, sensitivity, and specificity). Conclusion: AI-supported systems show potential performance increases in detecting and quantifying steatosis, NASH, and liver fibrosis in NAFLD patients. Before real-world implementation, prospective studies with direct comparisons of AI-assisted modalities and conventional techniques are necessary.

## 1. Introduction

### 1.1. Background

Non-alcoholic Fatty Liver Disease (NAFLD) is a group of disorders caused by a build-up of fat in the liver. The disease is most common in overweight or obese people [1]. One of the most common chronic liver disease in the world is NAFLD, affecting between 25% and 30% of the adult population [1,2]. High liver fat levels are linked to a higher risk of significant health issues such as diabetes, high blood pressure, cirrhosis, renal disease, and heart disease [3]. However, if diagnosed and treated early enough, NAFLD can be prevented from worsening, and the amount of fat in the liver can be reduced. Unfortunately, advanced liver disease and mortality due to NAFLD/NASH are expected to rise in Saudi Arabia (Figure 1), necessitating a strategy to slow the growth of the NAFLD population and minimize the liver disease burden [4]. The primary line of treatment for NAFLD and NASH is lifestyle changes, including eating habits and physical activity. A well-established therapy for NAFLD and NASH is weight loss, which has an unmistakable dose–response correlation between definite nutriments and fatty liver disease [5].

The progression of NAFLD involves four stages. Steatosis (simple fatty liver) is a harmless build-up of fat in the liver cells. A more severe form of NAFLD called NASH occurs when the liver becomes inflamed. A patient is diagnosed with fibrosis when persistent inflammation creates scar tissue around the liver and adjacent blood vessels, but the liver can still function normally. Cirrhosis, the most severe stage, develops after years of inflammation and causes the liver to shrink, scar, and lump; this damage is irreversible and can lead to liver failure (Figure 2). Typically, doctors divide the severity of a patient’s disease into four groups, normal, mild, intermediate, or severe, based on histological characteristics [6].

In most cases, abdominal ultrasonography is used to diagnose NAFLD [7]. Ultrasonography is a low-cost, safe, quick, and uncomplicated procedure in most healthcare settings [8]. To determine the severity of liver disease, non-invasive testing or liver biopsies are used. Hepatic steatosis, inflammation, and fibrosis are all assessed with a liver biopsy. However, a liver biopsy is an intrusive procedure that may result in a hemoperitoneum or hemothorax. Liver biopsy is also ineffective as a follow-up technique for liver disease due to its invasive nature [9]. Alternative NAFLD diagnostic approaches, such as clinical/laboratory scores, have been developed, although their efficacy has been questioned. The accuracy of magnetic resonance imaging proton density fat fraction (MRI-PDFF) is higher, but the cost and availability are limiting [10]. Compared to results in late-stage NAFLD, transient elastography and several biomarkers demonstrate better performance in the initial stages. Various diagnosis approaches are being combined with artificial intelligence (AI) to increase diagnostic performance [11].

AI is an area of computer science that tries to replicate humans’ learning and problem-solving abilities [12]. Expert systems made up of clearly stated rule-based algorithms [13] and machine learning (ML), where patterns are learned from data rather than human-designed rules [14], are the two primary forms of AI [15]. The three types of ML are unsupervised, supervised, and reinforcement. Unsupervised learning (e.g., k-means, clustering, and Gaussian mixtures) is used to find clusters based on shared characteristics. To teach mapping relationships between input-output (anticipated outcome) pairings, supervised learning (e.g., support vector machines (SVM), decision trees, and artificial neural networks (ANN)) are typically utilized. ANN are a computer analysis tool inspired by the neuroanatomy of the human brain. ANN have three layers: input, hidden, and output. The training phase allows neurons to optimize outcomes by weighting the incoming data. A convolutional neural network (CNN) is an image-based ML technology inspired by the human brain’s visual cortex. Convolutional, nonlinear, and pooling layers make up CNNs. A deep neural network (DNN) is made up of numerous filters that are applied to automatically find relevant aspects in the input data. Learning how to act in order to maximize a numerical reward signal is known as reinforcement learning. The learner must try each action to determine which produces the greatest reward. In the most fascinating and difficult situations, choices can influence not only the immediate reward but also the subsequent circumstance [16].

Numerous research has demonstrated the viability of combining AI with medical technologies to enhance outcomes, lower margins of error, and increase the standard of care. M.B. Jamshidi et al. found, for instance, in their review that various AI techniques can forecast the spread of COVID-19 [17]. In a distinct review, M.B. Jamshidi et al. also demonstrated how mobile applications are quickly becoming tools that provide a variety of services in the field of medical sciences, particularly in the COVID-19 era, and can effectively aid medical systems [18]. In fact, a substantial role for AI in battling COVID-19 [19] and creating vaccines in a short amount of time [20] has been demonstrated. Additionally, V. Srivastava et al.’s study improved the categorization accuracy for dermatoscopic images using AI approaches, increasing it to 96%. [21]. Another example was the work of D. Pal et al., where they showed that Attention UW-Net is the best for automatically segmenting and annotating organ x-rays due to its uniformity in prediction accuracy of segmentation masks [22]. Diverse types of AI have been used to address multiple liver problems using different interventions. Many studies on diagnosing, detecting, quantifying, and managing liver disorders such as fibrosis, focal lesions, carcinoma, NAFLD, and chronic illness have used AI to improve diverse diagnosis methodologies. For example, Piscaglia et al. employed ANN to predict the presence or absence of fibrosis in liver samples [23]. In addition, S-h Z et al. employed MRI scans and a ML technique to distinguish between malignant and benign liver cancers [24]. However, more studies are using AI to diagnose and determine the stage of NAFLD. For example, Vanderbeck et al. employed an SVM machine learning model to categorize digital normal and NAFLD biopsy images [25]. In addition, Naganawa et al. used a logistic regression (LR) machine learning model on non-contrast-enhanced computed tomography to predict stages of fatty liver disease [26].

Using ultrasonography with AI to detect and quantify NAFLD is a promising approach that has recently attracted researchers’ attention. In a review published in 2021, the domains of liver disease in which AI can be used are briefly discussed [27]. In addition, the use of ML for measuring liver fibrosis, forecasting hepatic decompensation, screening potential liver transplant recipients, and predicting post-transplant survival and complications was reported in a recent systematic review of AI in hepatology [28,29,30]. A comprehensive study also highlights basic technical knowledge about AI, such as traditional ML and deep learning (DL) algorithms, particularly CNNs, and their clinical applications in the medical imaging of liver diseases, such as detecting and evaluating focal liver lesions, facilitating treatment, and predicting liver treatment response [11,31]. Another recent systematic review detailed the use of AI in imaging modalities, digital pathology, and electronic health records to diagnose and stage NAFLD [32]. Furthermore, the performance of AI-assisted systems for the detection of NAFLD, NASH, and liver fibrosis is examined in a recent meta-analysis [33].

### 1.2. Research Problem and Aim

To the researchers’ best knowledge, no studies rigorously analyzed AI models for diagnosing and quantifying NAFLD using ultrasound (US) images or compared models regarding measures. Therefore, this systematic review is the first work of its kind in the literature. Many knowledge gaps have been identified and explored throughout this article. This study aims to examine all prior studies to determine the best accuracy, sensitivity, and specificity for diagnosing and quantifying NAFLD using ML, DL, or a combination of both.

## 2. Methods

### 2.1. Overview

A preliminary search and idea validation were conducted using the search terms “Artificial Intelligence” and “Ultrasound” and “Fatty liver” in PubMed and Google Scholar. When performing this step, systematic reviews and meta-analyses were found. These sources contained relevant papers to read to gain a deeper understanding of the topic and identify gaps to articulate the research question better. Because the previous reviews had different outcomes and populations, a systematic review of AI-powered ultrasonography were conducted to detect and quantify NAFLD. Figure 3 summarizes the method and tackles the research purpose.

### 2.2. Protocol and Registration

The protocol was designed by the author and reviewed and approved by the corresponding author. The protocol is registered at PROSPERO.

### 2.3. Search Sources

The bibliographic databases used in this study were PubMed, ACM Digital Library, IEEE Xplore, Scopus, and Google Scholar. The first one hundred Google Scholar results were sorted by relevancy to the search topic, returning many publications. The initial search was broad, based on terms in all fields across all databases. Because the number of keywords allowed in IEEE Xplore and Google Scholar was limited, shorter search strings were used compared to the other databases. Backward and forward reference list checking turned up a multitude of related studies in a variety of databases.

### 2.4. Search Terms

The search terms used in this review were related to the population (e.g., “fatty liver”, “hepatic steatosis”, “non-alcoholic fatty liver”, “metabolic associated fatty liver disease”, etc.), first intervention group (e.g., “Artificial Intelligent *”, “machine learning,” “deep learning”, “convolutional neural network *”, “artificial neural network *”, etc.), second intervention group (e.g., “ultrasound”, “ultrasonogr *”, “sonography”, etc.), and outcome (e.g., “detect *”, “quantif *”, “diagnos *”, etc.). Even though the comparator is usually essential for explaining the mode of comparison, no terminology related to it was included because a comparison based on comparators is outside the scope of this study. All comparators in the included studies were grouped into a single category termed “Control.”

### 2.5. Study Eligibility Criteria

All studies reported patients with hepatic steatosis, including NAFLD, NASH, acute fatty liver of pregnancy (AFLP), and other disease phases. Eligible interventions were those that fell under the umbrella of AI and were related to computer vision (CV), which was used to assess medical images. The type of medical image discussed in this review was a US, also known as sonography or ultrasonography. Interventions that identified and diagnosed fatty liver disease and its stages were considered eligible outcomes. Outcome accuracy, sensitivity, and specificity were used to assess performance. This review used no limitations such as age, gender, race, or publication date. Table 1 defines the inclusion and exclusion criteria.

### 2.6. Study Selection

Using PRISMA, studies were selected based on three processes: deleting duplicates, screening and assessing titles and abstracts of recovered research and index terms, and reading the contexts of the studies selected during the previous processes. The research selection was aided by a web-based systematic review tool known as Rayyan, speeding up the screening step [34]. Finally, the primary author completed all the processes and conducted data cross-checking among the studies’ extracted data to correct any probable errors.

### 2.7. Data Extraction and Synthesis

The primary author designed a data extraction form to collect specific data and parameters from the included studies. The form was a result of revising the parameters gathered in similar reviews and adding extra parameters needed to accomplish this review’s aim. The final extraction form was reviewed and approved by the corresponding author. Furthermore, data regarding the study (e.g., first author, year of publication, country of publication, publication type, etc.), population (e.g., age, gender, health status, sample size, etc.), interventions (e.g., imaging modalities, image types, image quality, AI branch, AI methods, validation methods, etc.), datasets (e.g., public/private, training data, testing data, augmented or not, etc.), and performance measures (e.g., accuracy, sensitivity, specificity, and Area Under the Curve) were collected by the author manually. In addition, as previously mentioned, manual data cross-checking among the studies’ extracted data was conducted by the author to correct any probable errors.

### 2.8. Risk of Bias in Individual Studies

Using the modified Quality Assessment of Diagnostic Accuracy Studies (QUADAS-2) was considered to evaluate the risk of bias in the studies, but all studies were included because of the limited number of studies available. A literature map was created to help review the literature for gaps and points of impact. According to Figure 4, the selected studies are mostly correlated since there are very few studies not in the citation loop. This result encourages including all the studies even though their quality is questionable. 

### 2.9. Data Checking Task

Data was validated through the cross-verification of more than two sources. Using triangulation, the consistency of findings was evaluated, increasing the chance of controlling any threats to result validity [35]. Collected parameters of included studies were cross-checked in front of the same parameter in different studies. A form that was designed by the author was used to record any discrepancy that was found. The next step was to revisit each paper and correctly extract the parameters that showed discrepancies according to the form.

## 3. Results

### 3.1. Search Results

Figure 5 depicts the search process and results. A total of 609 articles were found during a literature search. After eliminating 93 duplicates, 516 titles and abstracts were reviewed and 461 articles were rejected for the following reasons: irrelevant research (*n* = 398), incorrect intervention (*n* = 42), incorrect population (*n* = 16), and incorrect publication type (*n* = 5). Following that, 55 full-text articles were reviewed, with 23 being removed for the following reasons: incorrect research design (*n* = 1), incorrect population (*n* = 4), incorrect intervention (*n* = 13), irrelevant studies (*n* = 2), and articles written in languages other than English (*n* = 2). Finally, after completing the forward and backward review process, the total number of studies considered was forty-nine.

### 3.2. Description of the Included Studies

Results of the performance measures and assessments of each included study are summarized in Table 5. This table contains basic information about the whole study, including the study number in the reference, the type of categorization, the number of images, and the classifier. The AUC, sensitivity, accuracy, and specificity are then added. Each study’s findings are also summarized.

### 3.3. Study Characteristics

The highest percentage of studies were journal articles (*n* = 34, ≈69%), while most of the rest of the studies were conference proceedings (*n* = 14, ≈28.5%). However, one study was a dissertation. As shown in Table 2, studies were published between 1996 and 2021—however, there was a marked increase in published studies, especially DL studies, starting in 2014. When dividing the studies by year of publication, 22.4% were published in 2021, 8.16% in 2020, 10.2% in 2019, and approximately 24.5% between 2015 and 2018. More than a third of the studies were about DL (*n* = 17, 35%), and the rest discussed ML (*n* = 32, 56%). Finally, while India, the USA, China, and Portugal published the most significant number of studies, the USA, Taiwan, and Romania published the largest number of DL studies. 

### 3.4. Definition of Result Themes

#### 3.4.1. Evaluation of Modalities

The included studies used different modalities, probe frequencies, and settings. Table 3 shows the modalities and frequency ranges used in all the studies.

#### 3.4.2. Evaluation of Image Pre-Processing

Image pre-processing enhances an image’s quality by cleaning and organizing it for better feature extraction suitable for ML models. Usually, three steps are taken to prepare an image for further operations. First, images are resized. Second, images are de-noised by removing and smoothing noises. For example, speckle noise is removed using filters such as the Gaussian blur. Third, images are segmented by separating the background from the foreground. Many methods were used to process the US images in the included studies. The processing methods used include cropping [37,42,45,50,51,53,55,56,59,62,69,77,80,84], resizing [37,50,59,60,61,77,84], rotating [60,61,81], edge detection using “Active Snake Contour” [80] or “α-scale space derivative quadrature filters” [62], image squaring process [51], and contrast limited adaptive histogram equalization [36,37,51,54,81]. 

The included studies also used different methods to deal with the size of the images. Usually, a Region of Interest (ROI) resides around the center line and under an image’s near, focal, or far zones. It can be rectangular [48,74,75] or circular [59,81]. Typically, ROIs are chosen manually by an expert or automatically using methods like a self-organizing map [53]. In this research, some studies used the complete image as the ROI with a size of 434 *×* 636 pixels [60,61,63,64], 800 *×* 600 pixels [71], 960 *×* 720 pixels [51], or 1024 *×* 1024 pixels [39]. Other studies selected larger ROIs with 500 *×* 500 pixels [36,37,53]. Some studies selected medium-sized ROIs of 360 *×* 360 pixels [77], 334 *×* 334 pixels [62], 224 *×* 224 pixels [50], 128 *×* 128 pixels [38,40,41,42], or 100 *×* 100 pixels [79]. Finally, some studies selected small ROIs such as 23 *×* 23 pixels [78], 30 *×* 30 pixels [65,66], 32 *×* 32 pixels [47,48,49,52], 50 *×* 50 pixels [58,72], 64 *×* 64 pixels [55,56,57], or 75 *×* 75 pixels [84]. 

#### 3.4.3. Evaluation of Features

Features such as maximum probability [38,52,53,59,63,78,80], uniformity [38,52,53,59,60,63,70,80], entropy [37,38,49,52,59,63,66,70,71,73,74,76,80,82], contrast [38,49,52,59,63,65,66,70,71,73,74,76,78,80], run-length uniformity [38,52,70,80], attenuation [43,73,74,75,79], grey-level metrics [49,71,73,74,80,82], inverse difference [38,49,52,59,63,70,80], anisotropy, and pair correlation function [54] are examples of features that are repeated in studies of higher accuracy.

There are many methods used to extract features from US images. Among the extraction methods frequently used in studies with higher performance are the grey-level co-occurrence matrix method [38,52,54,58,63,71,76,79,80,83], grey-level difference statistics [48,49,66], grey-level run-length matrix [38,48,49,58], spatial grey-level dependence [47,48,49,53,65,66], wavelet packet transform [55,56,57], and first-order statistics [52,58,79].

Using only pertinent data and eliminating data noise, feature selection is a technique for lowering the variability of inputs put into a model. Choosing suitable characteristics for the AI model is based on the type of problem to be solved. Some feature selection methods often used in high-performance studies are locality-sensitive discriminant analysis [36,37], student’s *t*-test [41,53], Fisher’s discrimination ratio with Pearson’s correlation coefficient [65,66], sequential forward floating selection [79], marginal Fisher analysis with Wilcoxon signed-rank test [39], and Welch’s test [54].

#### 3.4.4. Cross-Validation and Data Splitting

Even though not all studies provided information about the methods used to manage data, most studies used cross-validation. Twelve studies implemented 10-fold cross-validation on the testing and training datasets [36,38,39,42,43,52,58,60,62,67,68,70], four studies used five-fold cross-validation [59,63,66,69], one study used four-fold cross-validation [77], two studies used three-fold cross-validation [41,54], one study used two-fold cross-validation [40], and one study used one-fold cross-validation [74]. In addition, four studies mentioned that a leave-one-out cross-validation (LOOCV) method was used [55,56,57,79]. 

Not all the included studies reported the rate of data splitting used for training and testing. Often, the percentage of data allocated for validation was also unclear. Table 4 shows approximate data splitting calculations. For most studies, the number of samples used for validation was assumed to be null since no numbers were explicitly declared for validation.

#### 3.4.5. Evaluation of Classification Models

An algorithm that performs classification is known as a classifier. The features of images that need to be classified significantly impact classifier performance. Numerous empirical studies have been conducted to compare classifier performance and identify the elements of images that affect classifier performance. 

Thirty-two included studies used ML models, and seventeen studies used DL models. In the ML studies, seven studies had their best performance using a SVM [52,55,56,57,58,60,67], four studies used a probabilistic neural network [37,39,54,76], and three studies used k-nearest neighbor [KNN] [48,49,82]. In addition, four studies used the Sugeno method [36], neural network method [47], or other fuzzy logic methods [74,78], and two studies used random forest [67,75]. Two studies used ensembled models. One study used three different classifiers: SVM, multi-layered perceptron neural net, and extreme gradient boost [70]. The other study used two classifiers: LR and SVM [68]. The rest of the studies used the following models to classify the images: binary logistic regression [BLR][71], adaptive boosting [67], Bayes [79], decision tree [41], single-layer perceptron network [73], regression tree model [43], single-layer feed-forward neural network [38], ANN [80], Levenberg–Marquardt back propagation neural network [40], and z-score [66].

DL studies used many models to classify US images. Seven studies used CNNs with variations in architecture (e.g., layers, branches, number of batches, pooling, convolution, etc.) [42,44,46,61,72,77,83]. The number of layers used ranged between three, as in [46], and twenty-two, as in [79]. In all seven studies, SoftMax was used for classification. In one study, Fourier CNNs with six layers were used as a classification model [64]. Two studies used Inception-v3 as a classification model [51,84], and two studies used VGG-16 to classify US images [50,69]. Five studies used a residual neural network (ResNet) with the following version variations: Inception-ResNet-v2 [63], multi-scale two-dimensional mid-fusion ResNet [62], ResNet-18 [45,51], and ResNet-50 v2 [81]. Finally, in one study, a SVM was used to classify images after a CNN was used to extract features [59].

#### 3.4.6. Explanation of the Performance Measure

Every machine learning problem may be divided into two categories: regression and classification [13]. For regression models, for example, measures such as Mean Absolute Error, Mean Squared Error, Root Mean Squared Error, and R2 are used to evaluate the performance. Classification models are also evaluated using measures such as Accuracy, Precision, Recall, F1-score, and AU-ROC. It is worth noting that the Metrics are distinct from the loss functions used to train a machine learning model, and they are typically distinguishable in the model’s parameters [85].

We will include the three key performance indicators reported in the majority of the included studies in this review: sensitivity, specificity, and accuracy. A few studies also reported F1-score and AUC; however, due to the scarcity of studies utilizing these measures, they will not be included in the review. The classification accuracy of a collection of measures is defined as how near they are to their actual value. Accuracy is calculated by dividing the number of correct predictions by the total number of predictions and multiplying the result by 100. A diagnostic test’s classification sensitivity measures how well it can identify true positives. Sensitivity is also known as Recall, Hit-Rate, and True Positive Rate. It may be computed by dividing the number of true positives by the total number of positives in the ground truth. Specificity, also known as Selectivity or True Negative Rate, assesses how successfully a test can identify true negatives. [86].

**Table 5 bioengineering-09-00748-t005:** Results of included studies.

Ref. No.	Patients Categories	Total no. of Images	AI Classifier	AUC	Sensitivity	Accuracy	Specificity	Main Findings
[36]	1- normal 2- abnormal	100	Fuzzy Sugeno (FS)	Unknown	100%	100%	100%	An automated diagnosis based on RT and DCT coefficients was used to classify a normal liver and a liver affected by fatty liver disease (FLD). Using only two features, the FS classifier presented the highest accuracy, sensitivity, and specificity, at 100%. Moreover, using just two elements, FLDI discriminated between normal and FLD.
[51]	1- Normal 2- Mild 3- Moderate 4- Severe	3200	Inception v3	Unknown	99.78%	99.91%	100%	The final neural network, SteatosisNet, used clipped L-K sections (using transfer learning and a second neural network) to categorize the severity of FLD. The experimental findings show that the suggested model may predict FLD effectively, comparable to the usual conclusions noted by medical professionals.
[69]	1- Normal 2- mild 3- moderate 4- severe	820	VGG-16	for B Modes Images: Mild = 0.71 Moderate = 0.75 Severe = 0.88 for Entropy Images: Mild = 0.68 Moderate = 0.85 Severe = 0.90	for B Modes Images: Mild = 73.18% Moderate = 63.25% Severe = 85.23% for Entropy Images: Mild = 64.10% Moderate = 70% Severe = 78.82%	for B Modes Images: Mild = 70% Moderate = 80% Severe = 97% for Entropy Images: Mild = 68% Moderate = 80% Severe = 83%	for B Modes Images: Mild = 60% Moderate = 74.82% Severe = 84.12% for Entropy Images: Mild = 70.16% Moderate = 86.54% Severe = 93.30%	When identifying mild and severe hepatic steatosis, there was no discernible difference between the VGG-16 model and entropy imaging. However, when it came to detecting moderate hepatic steatosis, ultrasonic entropy imaging performed better than the VGG-16 model. Interestingly, a physics-based analysis technique was as effective as DL and performed better at spotting mild to severe hepatic steatosis.
[44]	1- Fatty liver 2- Not Fatty liver	905	CNN	unknown	0.886	92.30%	95.30%	Diagnosing NAFLD by US was compared to radiologists’ performance. Cloud AutoML Vision Beta allowed the creation of custom models trained on uploaded images using a CNN pre-trained through transfer learning. The model accurately detected NAFLD on US.
[47]	1- normal 2- fatty 3- cirrhotic	150	Fuzzy neural network	unknown	unknown	Normal = 80% Fatty = 88% Cirrhosis = 80% Total = 82,67%	unknown	Through this work, proximity-based methods for building fuzzy neural classifiers in greater detail can be assessed, and more effective strategies for generating soft decisions can be learned.
[75]	1- Normal 2- Mild 3- Moderate 4- Severe	120	Random forest	Unknown	Unknown	90.84%	Unknown	Without using any features, RF had superior or comparable accuracy to SVM when classifying the severity of steatosis. In addition, human intra-observer and inter-observer agreement rates were outperformed by RF-based steatosis rating and SVM classification.
[37]	1- normal 2- FLD 3- cirrhosis	150	probabilistic neural network	0.98	96%	97.33%	100%	This work proposed a unique method for automatically distinguishing between a normal, FLD, and cirrhotic liver using US images. The technique combines CT, entropy features, and LSDA feature reduction. The suggested approach achieved high performance using a PNN classifier.
[45]	1- healthy 2- mild 3- moderate 4- severe	Unknown	ResNet-18	mild = 0.85, moderate = 0.90, severe = 0.93,	Unknown	Unknown	Unknown	The DL algorithm offers a trustworthy quantitative steatosis assessment across views and scanners in two multi-scanner cohorts. High diagnostic performance was achieved, matching or exceeding that of FibroScan.
[59]	1- normal 2- fatty	550	support vector machine	0.977	100%	96.30%	88.20%	This study used a steatosis level assessment utilizing B-mode US images via a CNN-based method. The method was effective and did not rely on an operator. Additionally, it performed better than both HI- and GLCM-based classifications.
[76]	1- normal 2- fatty	100	Probabilistic Neural Network	Unknown	Unknown	Normal = 85% Fatty = 87.25%	Unknown	To automatically classify and recognize fatty and normal liver, five joint statistical feature parameters [mean, variance, contrast, ASM, and entropy] retrieved from three approaches [grey histogram statistic, GLDS, and GLCM] achieved good results when utilized as the input of a PNN.
[77]	S0: H-MRS index < 3.12%, S1: H-MRS index > 3.12% and < 8.77% S2: H-MRS index > 8.77% and < 13.69% S3: H-MRS index > 13.69%	31,702	CNN	Unknown	Unknown	90%	Unknown	A high number of US images were used to train 5-layer CNNs. Results showed a good correlation with state-of-the-art magnetic resonance spectroscopy measurements.
[60]	1- Susceptible to FL [Steatosis > 5%] 2- normal people [<5%]	550	support vector machine	0.9999	97.20%	98.64%	100%	This method displayed and contrasted the outcomes of various DL algorithms based on how well they performed. The findings of this study demonstrated that the suggested pre-trained CNN could categorize US images of the liver as normal or fatty with excellent accuracy.
[78]	1- normal 2- diseased	Unknown	Fuzzy Classifier	Unknown	Unknown	100%	Unknown	This study identified how to automatically classify and recognize focal and diffuse liver diseases [including fatty and normal liver]. Advanced image processing methods such as MLPND and MI were used. Five features [contrast, cluster prominence, auto-correlation, cluster shade, and ASM] retrieved by the Haralick approaches achieved excellent results when utilized as the input of a fuzzy classifier.
[61]	1- Does not have steatosis 2- has steatosis	550	CNN	Unknown	Unknown	87.49%	Unknown	Using an 18-layer CNN with four convolutional layers resulted in an accuracy of 87.49%. Better image processing and dataset splitting techniques must be used for better results.
[73]	1- normal 2- fatty 3- cirrhotic	120	Single-Layer Perceptron Network	Unknown	For Cirrhotic: 91.7% For Fatty: 96.7%	Unknown	88.30%	Some features [mean grey level, first percentile, grey level co-occurrence matrix, contrast, entropy, correlation, ASM, attenuation and backscattering parameters, and scatterer separation distance] retrieved from GLCM approaches achieved good results for classifying fatty and cirrhotic liver when utilized as an input of a single-layer perceptron network with a functional link.
[41]	1- normal 2- fatty liver	100	Decision Tree	0.933	88.9%	93.3%	100%	This study had excellent performance results for classifying normal and fatty liver using three highly discriminatory noteworthy features [texture homogeneity, texture run percentage, and short-run emphasis] to train and build two supervised-learning-based classifiers [decision tree].
[79]	1- healthy 2- steatotic	75	Bayes	Unknown	normal = 95.83% steatosis = 85.71%	93.54%	Unknown	This study’s key finding was that the AR coefficients obtained from a multi-scale Haar wavelet decomposition were relevant for classifying hepatic steatosis using US images. The results of global and local assessments of liver tissue defined by the Bayes factor can give doctors valuable information about the classification’s confidence and the classification itself.
[42]	1- normal 2- abnormal		CNN	1	100%	100%	100%	To reduce dimensionality and DL network speed without raising computational expenses, the system in this study used the inception model. First, the background of the original liver images was removed from the optimized images by stripping the border. When removing 15% of the background, the findings showed remarkable accuracy.
[84]	1- normal patient 2- fatty liver patient	629	Inception-v3	0.93	89.90%	93.23%	96.60%	This study used the Inception-v3 to detect steatosis and classify normal and fatty liver images, yielding an excellent test performance.
[48]	1- normal 2- fatty 3- cirrhosis 4- hepatoma	unknown	K-nearest neighbour	Unknown	Unknown	80%	Unknown	Using GLDS, RUNL, SGLDM, and FDTA algorithms, this study used a method created for computer-assisted liver tissue characterization. It was anticipated that it would be challenging to distinguish cirrhosis, fatty, and diffused diseases from normal, but the preliminary outcomes seemed incredibly good.
[53]	1- normal 2- fatty	100	Self Organising Map	Unknown	Unknown	Unknown	Unknown	This study found representative feature vectors using a one-dimensional self-organizing map [SOM]. The most distinctive components were “maximum probability” and “uniformity.” The plots for normal and fatty liver superimposed images indicate distinct groups with little to no overlap.
[72]	1- normal 2- mild 3- moderate 4- severe	852	CNN	0.958	Unknown	95.45%	Unknown	In the NAFLD diagnosis stages, envelope signal and grayscale values were essential components of this study. However CNN showed the highest sensitivity and specificity when determining the severity of NAFLD. In addition, the deep-learning index had the best diagnostic performance in differentiating between mild and severe NAFLD (AUC = 0.958).
[55]	1- normal 2- fatty 3- heterogeneous	88	SVM	Unknown	Heterogeneous= 100% Fatty= 93.3% Normal= 86.4%	91%	Unknown	In this study, a suggested algorithm distinguished between normal, fatty, and heterogeneous liver images. Two steps make up the proposed algorithm’s operation. Without the aid of a medical specialist, the first stage automatically chooses a few ROIs from a liver US image. Then, the wavelet packet transform [WPT] was applied to chosen ROIs as a multi-scale texture analyser to extract some statistical features. A hierarchical binary classification method with an SVM classifier was used in the second stage.
[49]	1- fatty 2- cirrhosis 3- normal	90	K-Nearest Neighbour	Unknown	Unknown	82.2%	Unknown	The FDTA and the SGLDM were the texture analysis methods employed in this study. On three sets of liver US images—fatty, cirrhotic, and normal—algorithms were used. A 32 × 32 pixel ROI was used to extract textural features. A kNN classifier was used to categorize the results. Together, the FDTA and SGLDM provided an accuracy of 82.2%.
[74]	1- normal 2- fatty 3- cirrhotic	140	Fuzzy logic	unknown	cirrhosis = 94% Fatty = 96%	Unknown	92%	In this study, features such as the mean grey level, 10^th^ percentile, contrast, ASM, entropy, correlation, attenuation, and speckle separation, produced good results when used as the input of fuzzy logic to build an automated categorization of cirrhosis, fatty, and normal liver. The findings of this research demonstrated the potential benefit of taking fuzzy reasoning into account during the “quantitative tissue characterization” of diffused liver diseases.
[80]	1- Normal liver 2- abnormal liver [cirrhosis, fatty liver, hepatomegaly]	60	ANN	Unknown	95%	95%	Unknown	In this study, the feature set employed, training samples chosen, and the classifier’s ability to learn from the training examples all impacted how accurate the ANN classifier was. A comparison strategy indicated that the GLRLM and the mixed-feature set demonstrated high accuracy during both training and testing.
[62]	1- normal 2- diseased	550	multi-scale two-dimensional mid-fusion residual neural network	Unknown	abnormal: 95.37% normal: 82.40%	91.31%	abnormal: 92.42% normal: 88.99%	The study proposed a multi-scale two-dimensional mid-fusion residual neural network for improving NAFLD classification from US data and a GAN-based network for image synthesis to enlarge the training dataset (instead of using patch images). The study showed that fusing B-mode US features, local phase features, and radial symmetry features at a mid-stage outperform early and late fusion, which indicates a strong correlation among unique features obtained after convolution operation.
[50]	1- normal 2- abnormal	157	VGG16	0.96	95%	90.60%	85%	The study suggested DL, transfer learning, and fine-tuning as methods for identifying fatty liver in US pictures with comparable performance to other similar studies.
[56]	1- Normal 2- Fatty 3- Heterogeneous	88	ν-linear support vector	Unknown	Fatty = 93.3% Normal = 97.4% Heterogeneous = 94.7%	95.40%	Unknown	The diagnosis of FLD and heterogeneous liver utilizing textural analysis of liver US images is a unique method presented in this research. First, a WPT was used to examine the ROI, and from each of the WPT sub-images, several statistical features were collected (median, standard deviation, and interquartile range). The classification was then performed using a “v-linear support vector” classifier. The suggested approach provided an overall accuracy of approximately 95%, demonstrating the system’s effectiveness.
[81]	1- Normal 2- Mild 3- Moderate 4- Severe	21,855	ResNet-50 v2	Normal =0985 Mild = 0.974 Moderate = 0.971 Severe = 0.981	0.838	0.841	0.948	In this study, ResNet-50 v2 was trained and evaluated on many images and, as a result, performed relatively well compared to invasive diagnostic techniques for fatty liver.
[54]	1- normal 2- fatty	340	probabilistic neural network	Unknown	100%	99%	97%	This study revealed that it is possible to differentiate between normal and fatty liver images using the anisotropy feature supplied to PNN.
[82]	1- normal 2- steatosis 3- hepatitis 4- cirrhosis	Unknown	k-nearest neighbour	Unknown	Unknown	normal = 86% steatosis = 90% hepatitis = 85% cirrhosis = 50%	Unknown	In this study, to automatically classify and recognize diffused liver diseases, three features (for steatosis: mean grey value, and for cirrhosis: mean grey value, texture energy, entropy) were retrieved from the GLCM approach. The approach achieved satisfactory results (except for cirrhosis) when utilized as a kNN input.
[57]	1- normal 2- fatty 3- heterogeneous	88	support vector machine	Unknown	98.84%	98.86%	Unknown	In this work, feature fusion techniques were used to create a computer-aided diagnostic system for the hierarchical classification of normal, fatty, and heterogeneous liver US images. The prominent features of the parallel- and serial-fused feature spaces were chosen after features were extracted (energy, energy deviation, median, standard deviation, and interquartile range). Using the LOOCV technique and the SVM classifier, serial and parallel feature fusion modes, achieved maximum classification accuracies of 100% and 98.86%, respectively.
[71]	1- healthy 2- diseased	16,551	A Binary Logistic Regression (BLR)	0.986	95.45%	95.74%	96.00%	According to the findings, US images are more dependable than CT imaging for detecting hepatic steatosis. In addition, when ten features from a co-occurrence matrix were loaded into a BLR, it performed pretty well at differentiating between healthy and diseased fatty liver.
[68]	1- normal 2- fatty 3- cancerous	114	logistic regression + support vector machine	Normal = 0.959 fatty = 0.956 cancer = 0.985	Unknown	87.50%	Unknown	The goal of this study was to examine the performance of a hybrid classifier (SVM and LR) in the diagnosis of liver steatosis utilizing a variety of US image features that were retrieved, including mean, SD, arithmetic mean, geometric mean, and skew.
[63]	1- malignant fatty livers 2- benign fatty livers	550	Inception-ResNet-v2	0.992	Unknown	98.50%	92%	The study results showed that the Inception-ResNet-v2 architecture-based model is more helpful in classifying medical images. In addition, the study showed that it performs better than classical methods regarding accuracy and AUC.
[38]	1- normal 2- abnormal	63	single layer feed forward neural network [SLFFNN]	0.97	97.59%	96.75%	unknown	This study built an extreme learning machine (ELM) on a single-layer feed-forward neural network. Only hidden-to-output weights were taught, and input-to-hidden layer weights were created randomly to reduce computing costs. As a result, the results were more accurate with fewer features.
[58]	1- Normal 2-Steatotic	177	Support Vector Machine	0.88	Unknown	79.77%	Unknown	The results of this study indicated that the SVM was the most applicable for the discrimination of pathologic tissues in clinical practice, having better performance than the kNN and ANN.
[65]	1- Fatty liver 2- Normal liver	30	Fisher’s linear discriminative analysis	Unknown	100%	92%	Unknown	This paper suggested a quantitative metric for the characterisation of the liver based on texture analysis. This process was motivated by the visual criteria used by radiologists.
[67]	1- fibrosis 2- activity 3- steatosis	144	adaptive boosting, random forest, support vector machine	adaptive boosting = 085 random forest = 085 support vector machine = 0.85	adaptive boosting = 87.5% random forest = 87,5% support vector machine = 93.8%	adaptive boosting = 85% random forest = 85% support vector machine = 85%	adaptive boosting = 76.9% random forest = 76.9% support vector machine = 69.2%	In this study, three different image types were utilized to extract features, and the analysis and classification results were satisfactory.
[66]	1- Fatty liver 2- Normal liver	180	Z-score	Unknown	100%	95%	90%	In this study, the best textural characteristics for classifying livers were found. A novel classification approach employing information fusion was suggested. It consisted of a linear combination of features weighted according to how well they could separate classes.
[43]	1- Normal 2- Mild 3- Moderate 4- Severe	unknown	regression tree model	0.93	87.50%	90%	92.86%	This study suggested that an existing learning-based model may perform well by combining US and shear wave features (shear wave attenuation, shear wave absorption, elasticity, dispersion slope, and echo attenuation). Furthermore, it supports that the target tissue may be identified and distinguished from other targets in the high-dimensional space established by the suggested ultrasonic parameter set.
[39]	1- Normal 2- Fatty	100	probabilistic neural network	0.9674	96%	98%	100%	GIST descriptors were used in this study to extract features. A marginal fisher analysis (MFA) data reduction method reduced many elements to the top seventeen. The Wilcoxon signed-rank test was used to create effective and reliable classifiers to rank a set of characteristics. Using eighteen features, the proposed approach identified all normal classes as normal (specificity was 100%). To train the classifiers, 10-fold stratified cross-validation was employed. The PNN classifier produced results with the highest classification accuracy of 98%, sensitivity of 96%, specificity, and PPV of 100%.
[40]	1- normal 2-abnormal [fatty liver, hepatomegaly, cirrhosis]	62	Levenberg–Marquardt back propagation neural network	Unknown	0.9808	0.9758	0.9722	The proposed system was successfully able to detect and classify the FLD.
[46]	1- S0 (none), 2- S1 (mild), 3- S2 (moderate), 4- S3 (severed)	300	Deep Convolutional Neural Network	Unknown	Unknown	87.50%	Unknown	The outcomes demonstrate the power of deep convolutional neural networks (DCNN) and the higher information richness of RF data over B-mode for NAFLD staging.
[52]	1- Normal 2- Mild 3- Moderate 4- Severe	53	support vector machine	Unknown	Unknown	85.4%	Unknown	In this study, classifying normal, mild, moderate, and severe liver images was objectified using medical domain knowledge to diagnose the severity of fatty liver images. Findings demonstrated that the classification accuracy for a given feature category, such as run-length matrix (RLM), may be improved by appending feature sets.
[83]	1- normal liver 2- low-grade fatty liver 3- moderate grade fatty liver 4- severe fatty liver	500	convolution neural network	Unknown	83%	90%	95%	The study covered the impact of network width on a model. The study found that correctly expanding the network model’s width increased the model’s accuracy. “Skip connection” expedites network convergence while preserving the image’s original features.
[70]	1- normal 2- positive	744	Support Vector Machine + Multi-Layered Perceptron Neural Net + Extreme Gradient Boost	Training set = 0.978 Testing set = 0.951 Validation set = 0.937	Unknown	Unknown	Unknown	In this study, twenty-eight features were retrieved from US images using Mazda software after wavelet transforms were applied to process images. Features were used to distinguish between a healthy liver and NAFLD in paediatric individuals using an ML-based predictive analytic model [ensemble model]. The model did well in classification.
[64]	1- Fatty 2- Normal	550	Fourier Convolutional Neural Networks	Unknown	Unknown	84.40%	Unknown	This study suggested that to increase the classification speed of medical images, Fourier layers are more feasible.

## 4. Discussion

Even though the first empirical research on this review topic was published in 1996, it must be acknowledged that few in-depth studies on this topic have had conclusive findings. For instance, Figure 6 shows the differentiation of the studies based on three factors. Regarding AI methods, some studies used ML, while others used DL. Regarding outputs, some studies detected disease while others quantified it. This quantification focused on fatty liver disease or incorporated morbidity stages. If the protocol of this review defined studies that used DL to quantify fatty liver disease and its later morbidity phases, the number of studies would be relatively small.

Figure 7 depicts the process steps followed in all the studies. Most studies used the same method to design prediction models using US images of the liver. The process consisted of collecting images, processing images to extract features, processing features in the classifier, and making a prediction. However, other studies used varied approaches, potentially resulting in disparities in performance.

A turning point in the computerized automated detection of fatty liver disease was reached when Michal Byra conducted a study in 2018 [59]. In Byra’s work, hepatic feature extraction was done using Inception-ResNet-v2 architecture. On US scans obtained from fifty-five patients, quantitative validation was conducted (38 fatty livers, 17 healthy livers). An SVM classifier was used to classify the retrieved features, and the reported mean accuracy was 96.3 percent. Since 2018, most studies have used Byra’s work as a benchmark against which to compare their findings. Five studies reviewed in this work used Byra’s dataset [60,61,62,63,64]. Furthermore, it is crucial to emphasize the significance of two recent studies conducted in Taiwan, where a considerable number of images from various imaging modalities were used in each study [45,81]. 

The studies included in this review show that combining AI with US image analysis can reduce human-related mistakes and enhance overall performance. The studies also demonstrate the capacity of AI-integrated approaches to detect early-stage steatosis. The studies shows impressive AI-assisted US performance with great sensitivity, specificity, Positive Predictive Value (PPV), Negative Predictive Value (NPV), and accuracy. 

Although most studies collected liver US images using Philips, Siemens, or GE modalities, it does not appear that the type of modality is related to the performance of the classification model. The modality settings and the frequency of the probe used in each study may also be observed similarly. Although some studies contend that utilizing a high frequency, such as 40 MHz, will result in greater classification performance (accuracy of 95%) [43], a comparative study using a low frequency, 5 MHz, scored the same performance [56]. It is worth mentioning that most studies used a frequency probe of 3.5 MHz.

Even though studies in this review processed images and chose ROIs in numerous ways, most of the studies unequivocally agreed on the importance of conducting these two processes. Regarding image pre-processing and ROI selection, no discernible patterns can be reported. Studies consistently standardized image size, removed irrelevant details, attempted to choose ROIs near images’ central lines, and avoided anomalies such as vessels and bile stores. Two studies successfully selected ROIs automatically. Ribeiro et al. based their approach on the decomposition of liver parenchyma US images in two fields—the speckled image holding textural information and the de-speckled image comprised of liver intensity and anatomical data [79]. Owjimehr et al.’s ROI selection process included three steps. First, images were divided into blocks and overlapped repeatedly. Second, sections of a specific size were chosen from the middle of each block. Finally, a linear support vector classifier was used to select the best ROIs from all those formed [55].

Most studies that used supervised ML reported selecting features for classification. While it is well known that the number of elements positively effects accuracy, feature types seem to have the most impact on accuracy. For example, a study that used 325 features to detect whether a liver was normal or steatotic scored an overall accuracy of 79.77% [51]. In comparison, another study that used only five features for the same purpose scored 100% accuracy [36]. The same can be seen with 636 features used and 85.4% accuracy in [69] and five features used and 87.5% accuracy in [44]. Likewise, one study selected 156 features and scored 85% accuracy [47], while another used only nine features and achieved 90.84% accuracy [75].

Regarding data splitting, different studies used different methodologies. However, using 80 to 90% of the data as a training dataset and 10% to 20% as a testing dataset were the most common splitting percentages. Furthermore, many studies used the 10-fold cross-validation method to validate data. 

Studies have proved the efficiency of using DL classifiers to classify images, and medical images are no exception. DL algorithms use data to learn high-level features. This is a distinguishing feature of DL and a significant advancement over classical ML. As a result, DL minimizes the need to create a new feature extractor for each challenge. However, when it comes to US images for NAFLD, studies show that a neural network AI outperforms a non-neural network AI. More research and quantitative analyses are needed to accurately identify a superior algorithm among the ones described in this review.

Finally, the included studies reported many challenges and opportunities for improvement. The following is a list of the most important obstacles to overcome in future research:to overcome problems that currently exist in some classifiers, such as speckle noise, semantic gap, computational time, dimensionality reduction, and accuracy of images retrieved from a large dataset;to examine the effect of every parameter to improve the performance of the model;to use a more extensive dataset acquired by different operators from different patients;to consider a multipolar hospital;to consider more diseases stages;to use more advanced techniques to improve images before analysis;to automate all steps as much as possible;to examine more sophisticated features; andto implement classification models in the hardware and transfer the technology to a clinical setting.

### 4.1. Clinical Implications

Although the studies did not elaborate on the clinical implications of suggested solutions and models, several studies attempted to discover the most appropriate models for healthcare settings, even if it meant sacrificing the quality of results. For example, one study attempted to reduce computation time and enhance speed by employing Fourier layers to standardize the modern technology in clinical settings [64]. Furthermore, the results in [58] suggested that SVM was best suited for differentiating diseased tissue in clinical practice, outperforming KNN and ANN.

Despite the above, several studies suggested that having a reliable technology that can be adapted to healthcare settings has clinical implications. The clinical implications were stated as secondary results, a conclusion, or recommended future works. The clinical implications are as follows:US powered by AI can be used to integrate an index in place of the H-MRS index of the biopsy method, which is invasive, expensive, scarcely available, and unsettling for patients [36,44,77]. US powered by AI also lessens the workload and the need for biopsy since it is considered a preliminary test for selecting patients eligible for biopsy [39,81].In the future, DL might be used to quantify NAFLD with the combined use of pathologic and laboratory tests [72].Non-invasive techniques with excellent accuracy would be superior evaluation tools to biopsy [45,55].Using AI on a 2D US liver scan will decrease subjectivity in diagnosing and increase reliability [37,45,57,72].Given the rising incidence of NAFLD and the potential for permanent hepatic damage, early recognition of NAFLD and cirrhosis is essential for doctors to be able to advise on appropriate therapies to stop the onset of HCC and its associated consequences [37,44].The accuracy of NAFLD detection with ultrasonography can be enhanced with the development of computer-aided diagnostic technology, especially for those less trained or operating in distant locations [50,51].Various image processing techniques will improve image quality, enhancing clinical interpretations and grading performance [41,53,68,69,78].Some methods provide a fully automated solution that will assist in determining the advantages of telehealth [40,56].Future US devices will include functionalities for tissue analysis that are easier to implement in hardware [73].

Potential readers of this work might include healthcare practitioners and computer scientists to promote awareness of the importance of collaboration between the two fields. Healthcare professionals, for example, can supply the necessary dataset for fatty liver to computer scientists, who can then run additional tests and act on clinical validation and feedback.

### 4.2. Strengths

To the best of the researchers’ knowledge, this is the first review to investigate all AI strategies used to automate NAFLD detection and quantification. The search was sensitive and accurate since the most prominent health and information technology databases were searched using a well-developed search query and backward and forward reference list checking. Because this research does not focus on individual AI branches or stages, it may be considered comprehensive. As a result, the study presents a comprehensive view of AI’s function in monitoring fatty liver using US images. On the one hand, the review may be deemed high-quality since well-recommended criteria were followed during the creating, implementing, and reporting processes. On the other hand, it is possible to expand on this work.

### 4.3. Limitations

Despite data cross-checking between studies being used to fill any gaps in the gathered data, it could be a limitation that only one person carried out the review was. Another drawback is that the search conducted for this study only covered English studies. As a result, studies written in other languages were omitted. Finally, the analysis presented in this research is qualitative. Therefore, it would be preferable to contribute additional value to the topic by conducting a metadata analysis on some of the included papers.

## 5. Conclusions and Future Work

Over time, efforts to detect and classify fatty liver disease and its accompanying clinical stages more accurately than humans have increased. Most of the effort has been devoted to extracting features from processed images and employing these features to complete the task. Using ANNs, whether for extracting features or classifying, represents a significant step in the right direction.

For potential future work, more effort needs to be placed into creating models that tackle challenges and performing randomized clinical trials on more significant numbers of patients. The findings will help in the future development of explainable AI. Furthermore, more efforts must be devoted to processing images and extracting features to determine the most accurate stages of the images, taking into account the structural differences between the images. In addition, comparing computational complexity/power and classification accuracy should be considered a strategy for comparing DL methods with ML methods. This will lead to a more advantageous selection to detect and quantify NAFLD using US images.

As AI has received a lot of attention regarding its utilization in the healthcare sector, this study emphasizes the application of AI in fatty liver diagnosis and future problems. Our findings pave the way for computer scientists to focus on the use of AI in the diagnosis of fatty liver, particularly in the early stages of the illness, which is difficult to identify, especially for junior non-expert doctors. As a result, AI applications are critical in this domain to overcome challenges and avoid human mistake.

## Figures and Tables

**Figure 1 bioengineering-09-00748-f001:**
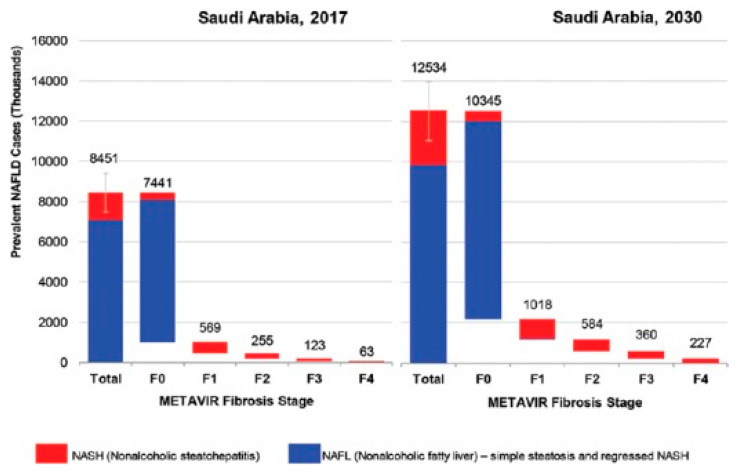
Distribution of NAFLD population by fibrosis stage—2017 and 2030, adapted from [4].

**Figure 2 bioengineering-09-00748-f002:**
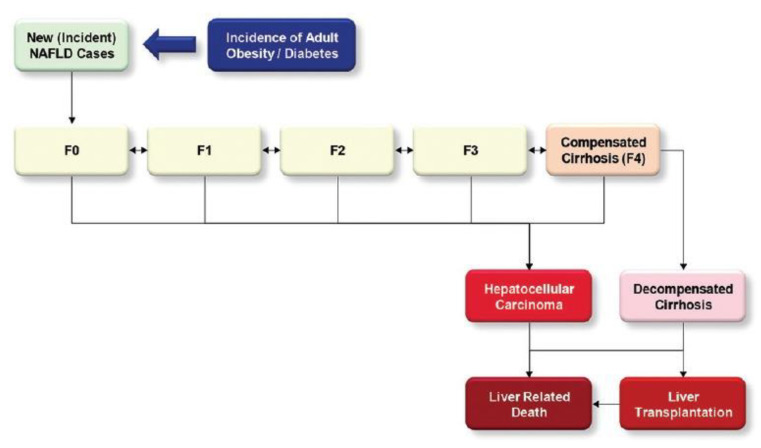
Example of NAFLD progression model [liver fibrosis can be divided into four stages (F1–4) as follows: F0—no fibrosis; F1—portal fibrosis without septa; F2—portal fibrosis and few septa; F3—numerous septa without cirrhosis; F4—cirrhosis, adapted from [4].

**Figure 3 bioengineering-09-00748-f003:**
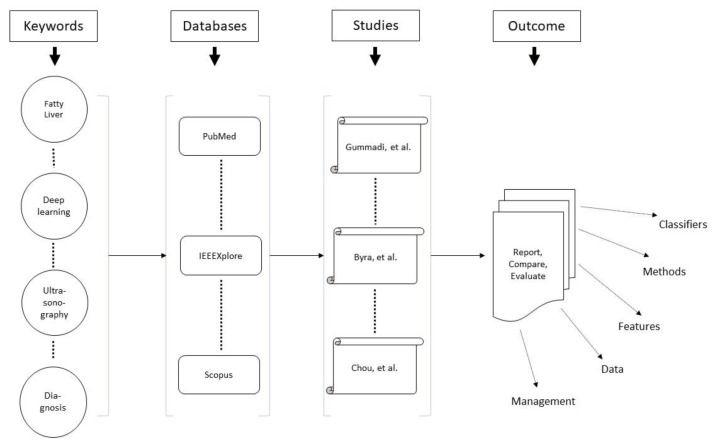
Method Summary.

**Figure 4 bioengineering-09-00748-f004:**
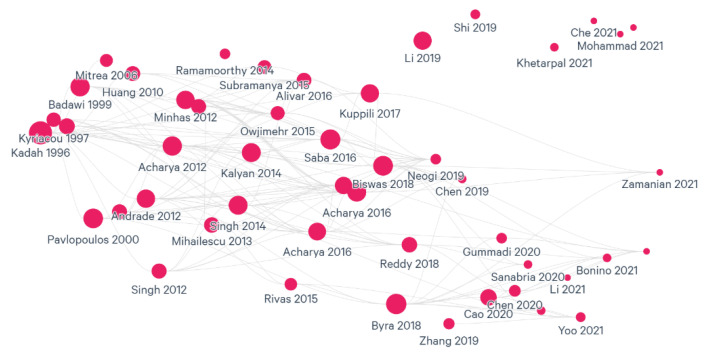
Selected studies’ relevance and recommended papers.

**Figure 5 bioengineering-09-00748-f005:**
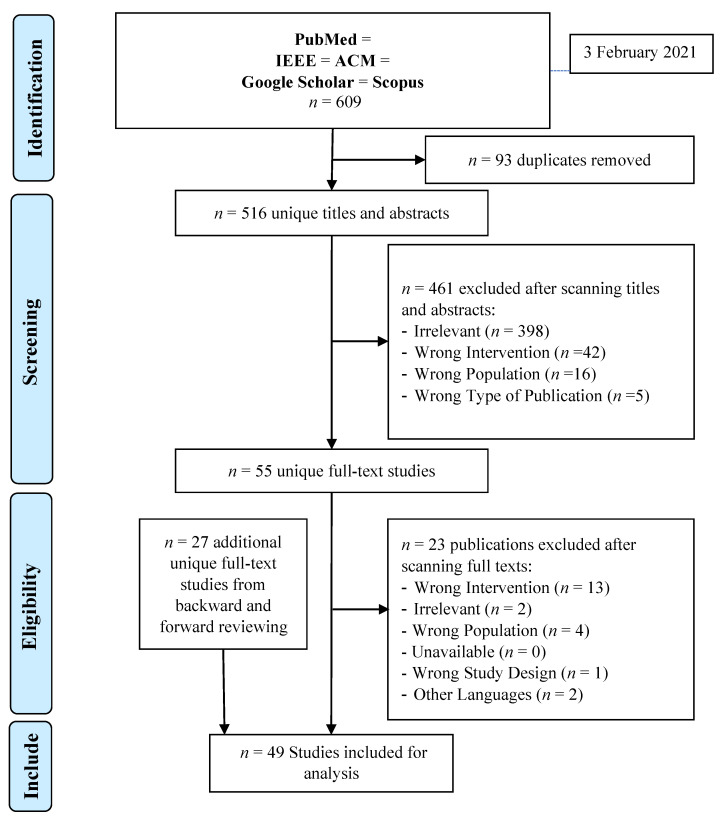
PRISMA diagram for study selection.

**Figure 6 bioengineering-09-00748-f006:**
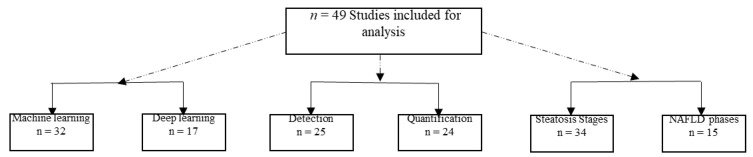
Separation of studies based on three factors.

**Figure 7 bioengineering-09-00748-f007:**
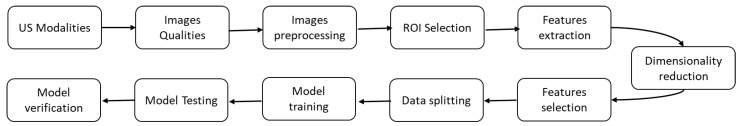
Methodology used to design models.

**Table 1 bioengineering-09-00748-t001:** Inclusion and Exclusion Criteria.

Parameters	Inclusion Criteria	Exclusion Criteria
Population	Patients with Hepatic Steatosis (NAFLD) and developed stages.	- Alcoholic fatty liver disease (AFLD) includes simple AFLD and alcoholic steatohepatitis (ASH). - Patients suffering from liver illnesses other than steatosis (e.g.,: tumors).
Intervention	AI that used ultrasound images to detect and quantify hepatic steatosis.	Non-AI-based technologies and AI technologies used other types of imaging (MRI, X-ray, etc.)
Comparator	N/A	N/A
Outcome	Detection and quantification of hepatic steatosis.	Any other outcome that is not mentioned in the inclusion criteria
Performance measures	The metrics to be measured are accuracy, sensitivity (recall), specificity, or AUC.	Any other measures that are not mentioned in the inclusion criteria
Study Type	Peer-reviewed articles, theses, dissertations, and reports.	Reviews, conference abstracts, and proposals.
Study Design	Empirical studies.	Any other study design that is not mentioned in the inclusion criteria
Study language	English	Studies written in any language other than English.
Study time frame	No limitation	No limitation
Age, Gender, Ethnicity	No limitation	No limitation

**Table 2 bioengineering-09-00748-t002:** Number of studies sorted by AI type.

Country	Total Number of Studies	Studies with Deep Learning Classifying Approach	Studies with Machine Learning Classifying Approach	Publication Period *
India	8 (≈16%)	1	7	2007 → 2019
USA	7 (≈16%)	5	2	1996 → 2021
China	5 (≈10%)	2	3	2010 → 2020
Portugal	5 (≈10%)	1	4	2012 ← 2018
Romania	4 (≈8%)	2	2	2006 → 2021
Taiwan	4 (≈8%)	3	1	2019 → 2021
Greece	3 (≈6%)	0	3	1997 ← 2000
Iran	3 (≈6%)	0	3	2015 ← 2021
Malaysia	3 (≈6%)	0	3	2016
Italy	2 (≈4%)	1	1	2016–2021
Egypt	1 (≈2%)	0	1	1999
Korea	1 (≈2%)	1	0	2021
Pakistan	1 (≈2%)	0	1	2012
Poland	1 (≈2%)	1	0	2018
Venezuela	1 (≈2%)	0	1	2015
Total	49	17 (≈35%)	32 (≈56%)	1996 → 2021

*: The arrow indicates an increase or a decrease in studies over time.

**Table 3 bioengineering-09-00748-t003:** Modalities and frequencies used.

Modality Manufacturer	Modalities Model	Average Frequencies in MHz	Studies Reference Number
Philips	CX 50	3.5	[36,37,38,39,40]
	CX c50	3	[41,42]
	EPIQ	40	[43]
	EPIQ 7G	3	[44,45,46]
	HD15	-	[45]
	IU22	-	[44,45]
Siemens	ACUSON 128XP/10	3.5	[47,48,49]
	ACUSON S1000	-	[50]
	ACUSON S2000	-	[45]
	ACUSON sequoia 512	4.5	[51]
	ACUSON X300	-	[52]
	Sonoline Versa Plus	3.5	[53,54]
Toshiba	SSA-700A	-	[45]
	SSA 550	5	[55,56,57]
	Xario	-	[45]
	TUS-A300	-	[45]
GE	Logic E9	4	[44,45,58]
	Vivid E9	2.5	[59,60,61,62,63,64]
	Voluson 730 Pro	3.5	[65,66]
	Logic S8	-	[45]
Canon	Aplio 500	3.5	[44,67]
	i800	-	[44]
Hitachi	Avius	-	[45]
	Preirus	-	[45]
CIRS	040GSE	-	[68]
Burlington	Terason 3000	3.5	[69]
Sonosite	M-Turbo	3	[70]
ESAOTE	MyLab 50	-	[71]
Mindray	Resona 7	5	[72]
KRETZ	SA 3200	4	[73,74]
Unknown	Unknown	-	[75,76,77,78,79,80,81,82,83]

**Table 4 bioengineering-09-00748-t004:** Data splitting rates.

In	AI learning Type	Ref. No.
≈50%, 50%, 0%	machine learning	[43,48,78]
≈56%, 44%, 0%	Machine learning	[82]
≈60%, 20%, 20%	Machine learning	[45]
≈60%, 40%, 0%	Deep learning	[77]
≈60%, 40%, 0%	machine learning	[81]
≈70%, 30%, 0%	Deep learning	[71]
≈70%, 30%, 0%	machine learning	[44,51]
≈75%, 25%, 0%	Deep learning	[72]
≈78%, 28%, 0%	Deep learning	[69]
≈79%, 21%, 0%	Deep learning	[70]
≈80%, 10%, 10%	machine learning	[84]
≈80%, 20%, 0%	Deep learning	[60,62]
≈80%, 20%, 0%	machine learning	[59,63]
≈80%, 9%, 11%	Deep learning	[61]
≈84%, 16%, 0%	Deep learning	[64]
≈88%, 12%, 0%	Deep learning	[83]
≈92%, 18%, 0%	Deep learning	[54]
≈94%, 6%, 0%	Deep learning	[46]

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
