# Peer review of "Artificial Intelligence for Detecting and Quantifying Fatty Liver in Ultrasound Images: A Systematic Review"

_bioengineering, 2022, doi:10.3390/bioengineering9120748_

Round 1
Reviewer 1 Report
I enjoyed reading this review manuscript. Authors summarized sufficiently for the AI models to detect and quantify fatty liver in US images using diverse range of literatures. My suggestion is it would be better to summarize the potential future research perspectives more precisely (for example, summarize in a single figure) in "Conclusion and future work".
Reviewer 2 Report
As mentioned in the abstract, this study explores how well various AI methods function and perform on ultrasound (US) images to diagnose and quantify non-alcoholic fatty liver disease. Five science bibliographic databases were searched, including PubMed, Association for Computing Machinery (ACM) Digital Library, Institute of Electrical and Electronics Engineers (IEEE) Xplore, Scopus, and Google Scholar. Only peer-reviewed English articles, conferences, theses, and book chapters were included. Data from studies were synthesized using narrative methodologies per Preferred Reporting Items for Systematic Reviews and Meta-Analyses (PRISMA) criteria. Results: Forty-nine studies were included in the systematic review. According to the qualitative analysis, AI significantly enhanced the diagnosis of NAFLD, Non-Alcoholic Steatohepatitis (NASH), and liver fibrosis. In addition, modalities, image acquisition, feature extraction and selection, data management, and classifiers were assessed and compared in terms of performance measures (i.e., accuracy, sensitivity, and specificity). Although detecting and quantifying fatty liver in ultrasound images based on AI seems interesting, there are some major issues that need to be addressed as follows:
1. The main purpose of this review needs to be explained clearly in the abstract and introduction. Going through the paper, an appropriate research question based on the problem cannot be observed. Authors need to clarify the potential readers to have a good understanding of the case of this topic, especially in terms of its scope and relevance.
2. It is better if the authors describe their methods using some extra charts and graphical works in this review paper. This could be a good alternative or better perspective to approach the research purpose.
3. While the authors used some references about applications of AI in medical images, it is highly recommended to use the more relevant references as follows:
a. A Review of the Potential of Artificial Intelligence Approaches to Forecasting COVID-19 Spreading
b. A Comprehensive Review of Radiology Smartphone Applications
4. It is highly recommended to add more details about the basic components (population, interventions, outcomes) of the study, and explain more about measurements validation, accuracy, and statistical significance. The conclusions need to be set based on accurate interpretations of the data.
5. Future directions?
Reviewer 3 Report
The review searched a big number of relevant literatures from five scientific literature databases using keywords such as "artificial intelligence", "ultrasound" and "fatty liver". It examined the prior studies to determine the best accuracy, sensitivity, and specificity for diagnosing and quantifying NAFLD using ML, DL, or a combination of both.
The authors used a narrative approach to synthesize the data and finally obtained the 49 references that best fit the topic. An overview of these references was given according to AI type, different modes used, frequency, cross-validation, and data splitting.
This review is of interest to other authors in this field. The authors finished the summarization of the published studies in this field from 1996 to 2021 and conducted some analysis of these studies. The review still has the problems:
1. The bar chart shown in Figure1 is not clear, and some of the bar charts are offset on the Y-axis. The annotated data is incorrect.
2. Thesis Evaluation of image pre-progressing part, spacing between words is too large, typesetting format has a problem. The same issue occurs in the Evaluation of features section.
3. Reference 67 in Table 5, where 'T67' uses an annotation that is inconsistent with the others.
4. The studies listed in Table 5 can be classified into categories to make it clearer. It is lack of a general summary of the references.
5. There is too much blank between pages 11 and 12.
6. The paper adopts horizontal layout from page 12 to 26, and the rest of the paper adopts vertical layout, suggesting a unified layout format.
7. References should be cited using "[ ]", not "( )".
8. Some references (e. g.24,28,36,37,40,43,46,47,50,52,53,58,59,60,61,62,71,
73,75,76,77,78) do not have a unified format, such as missing journal name and page number, journal name first letter case is not uniform, some page numbers are preceded by "p.", etc.
Reviewer 4 Report
* The need of Explanable AI part is missing. In what case doctors will use the AI need to discuss case by case basis. Impact of radiomics in co-clinical images is missing in this study *Further, in the introduction, what is the recent knowledge gap of the main literature that the author needs to write this research? What we have known and what we have not known? What is missing from current works? Please explain and give examples! * In terms of the knowledge gap, it will be best if the research challenge/knowledge gap could be stated in one article or more articles in the main literature (optional). in the main literature you can include the papers A median based quadrilateral local quantized ternary pattern technique for the classification of dermatoscopic images of skin cancer, and Attention UW-Net: A fully connected model for automatic segmentation and annotation of chest X-ray . Mention the usefulness of their method . * Research question must be explicitly stated in the introduction. Show how the main literature informs the formulation of your research question(s). * In terms of theoretical contribution, show the theoretical novelty that your work offers. How does this novelty distinguish your work from other similar works? * Practical Implications seem to be unclear? Please mention and make a reference! Further, please answer 1) for whom it is relevant? Who you are writing this manuscript for? - is there any formal definition for the role? 2) In which contexts? * As to practical implications, how do the findings help the health organizations? Please explain and give examples! Assure that any recommendation is clear and actionable for organizations. * As to practical implications, show how your recommendation is timely. * As to both theoretical contributions and practical implications, show how your research potentially influences health organizations and individuals to think, behave, or perform through multifaceted forms and channelsAuthor Response
Please see the attachment

Round 2
Reviewer 2 Report
All comments have been addressed
Reviewer 4 Report
Authors have revised the manuscript as per the comments. I have no further comments.